# Land Cover Types Drive the Surface Temperature for Upscaling Surface Urban Heat Islands with Daylight Images

Julien Radoux [1,*,†], Margot Dominique [1,†], Andrew Hartley [2], Céline Lamarche [1], Audric Bos [1] and Pierre Defourny [1]

1   Earth and Life Institute, Université Catholique de Louvain, 1348 Louvain-la-Neuve, Belgium; margot.dominique@uclouvain.be (M.D.); celine.lamarche@uclouvain.be (C.L.); audric.bos@uclouvain.be (A.B.); pierre.defourny@uclouvain.be (P.D.)
2   Met Office, Exeter EX1 3PB, UK; andrew.hartley@metoffice.gov.uk
*   Correspondence: julien.radoux@uclouvain.be; Tel.: +32-(0)10-479-257
†   These authors contributed equally to this work.

**Abstract**

The widespread availability and spatial coverage of land surface temperature (LST) estimates from space often result in LST being used as a proxy for near-surface air temperature in order to characterize the urban heat island (UHI) effect. High-spatial-resolution satellite-based LST estimates from sensors such as Landsat-8 provide the spatial and thematic details necessary to understand the potential effects of urban greening measures to mitigate the increased frequency and intensity of heatwaves that are projected to occur as a result of human-induced climate change. Here, we investigate the influence of land cover on Surface Urban Heat Island (SUHI) observations of LST using a technique to reduce the spatial spread of the per-pixel temperature observation. Additionally, using land cover-based linear mixture models, we downscale the surface temperature to a 2 m spatial resolution. We find a mean difference in LST, compared to the city average, of +8.94 °C (+/−1.87 °C at 95% CI) for built-up cover type, compared to a difference of −7.42 °C (+/−0.8 °C) for broadleaf trees. This highlights the potential benefits of creating urban green spaces for mitigating the UHI amplification of extreme heatwaves. Furthermore, we highlight the need for improved observations of night-time temperatures, e.g., from forthcoming missions such as TRISHNA, in order to fully capture the diurnal variability of land surface temperature and energy fluxes.

**Keywords:** urban heat island; thermal remote sensing; Landsat; land cover; heat waves; urban; unmixing

## 1. Introduction

Urban areas host 4.2 billion people, representing the majority of the global population. By 2050, urban population is expected to grow by an additional 2.5 billion, with 90% of this increase occurring in Asia and Africa [1]. Urbanization has exacerbated the effects of global warming in cities [2,3]. The combined effect of global warming and population growth in already warm regions significantly increases heat exposure, and the majority of the population exposed to heatwaves will live in urban centers [4].

Heatwaves are exacerbated by the urban heat island (UHI) effect, characterized by higher temperatures in urban areas compared to their rural surroundings [5]. UHIs are defined as areas inside cities where air temperatures are at least 2 °C warmer than nearby

rural areas [6]. The UHI impacts human comfort and health [7–10], biodiversity [7,11–16], and energy consumption [7,17–19].

The urban heat island (UHI) effect is primarily driven by changes in land use/land cover due to urbanization, which alters surface characteristics and urban morphology, disrupting the local energy balance [20]. Key contributors to urban microclimate changes include reduced evaporative and ventilative cooling, altered energy balances, and increased anthropogenic heat release [21]. The intensity and characteristics of UHIs vary with the day/night cycle [22,23], seasons [23,24], weather conditions [15,23–26], and geographical location [25]. Urban design elements also play a significant role, including morphology [22,27–31], building materials [23,28,32–36], vegetation [6,37–43], water bodies [39], and anthropogenic heat emission [32].

The UHI can be further characterized by the typical atmospheric layering over a city, forming three types of UHI: the Canopy Layer Heat Island (CLHI), the Boundary Layer Heat Island (BLHI), and the Surface Urban Heat Island (SUHI) [44]. SUHI refers to surface warming, while CLHI and BLHI refer to atmospheric temperature increases. CLHI involves the warming of the urban canopy layer, which is the air layer closest to the ground up to the average height of building roofs. BLHI refers to temperature increases in the urban boundary layer, which extends above the canopy layer [45,46]. During the day, BLHI can extend from building roofs up to 1 kilometer or more, while at night, it is restricted to a few hundred meters or less.

The intensity of UHIs, i.e., the temperature difference between urban areas and their rural surroundings, depends on two key variables: air temperature and land surface temperature (LST). Air temperature measurements are used to assess the Canopy Layer Heat Island (CLHI) and Boundary Layer Heat Island (BLHI), while LST is used for Surface Urban Heat Island (SUHI) evaluations [47]. LST, which represents the thermal radiance emitted from the land surface, influences air temperatures in the lower atmospheric layers, thereby affecting thermal comfort and the indoor climate of buildings [46,48]. By day, solar heating is strong, and the heat distribution favors heat storage over convection [44]. SUHI is therefore less connected to UHI in the morning than during the night.

Dense networks of air temperature measurements are crucial for accurately describing the spatial extent and intensity of CLHI and BLHI. However, these networks are not available in all cities worldwide [31,46]. Additionally, air temperature measurements are point-based, limiting their ability to delineate spatial contrasts. In contrast, LST is obtained via satellites using thermal infrared remote sensing, making it an inexpensive and globally available method [31,45]. Consequently, LST is often used as the sole indicator of UHIs, despite it only representing SUHI at a given time of the day. Although various thermal sensors exist, the current spatio-temporal resolution of thermal satellite missions is inadequate for effective heat island monitoring, necessitating the use of downscaling methods [49].

Visible or short infrared spectral bands, with their finer resolution, support LST downscaling. Many methods use the Normalized Difference Vegetation Index (NDVI) as a regression kernel [50]. However, combining spectral unmixing of land cover (LC) proportions with thermal unmixing outperforms models based on single indices [51]. Using multiple indices from very-high-resolution images, whether hyperspectral [52] or multispectral [53], also demonstrate improved accuracy.

Some relationships have been identified between LC proportions and LST using thermal images [54–58]. Artificial surfaces and bare soils exhibit the highest LSTs, while water bodies and vegetation have lower values [59]. In contrast, building height shows a weak correlation with LST [60]. However, despite promising findings, the determination coefficients of LC-based models remain moderate ($R^2$ from 0.48 to 0.81). This could be due to

imprecise estimates of the actual land cover proportions. On one hand, the relatively coarse resolution of the available LC maps used in these studies lead to mixed pixels in the heterogeneous urban areas (e.g., combining in the same pixel roofs, street pavement, terraces, trees, and gardens). On the other hand, all previous studies assumed that the information contained in a pixel was derived entirely from within its own footprint. In reality, however, a significant part of a pixel's spectral signal in a remotely sensed image originates from the surrounding areas [61–63]. This results from various factors, such as the detector design, atmospheric influences, and image resampling. These effects are collectively represented by the point spread function (PSF), which describes how a sensor responds to point sources of light.

Our study rigorously investigates the relationships between daylight remotely sensed LST values and LC class distribution. We account for signal spreading to analyze subpixel proportions, ensuring all ground surfaces contribute to the recorded value in each pixel like in several studies with optical data [63,64]. Using a very-high-resolution LC map to avoid mixed pixels, we aim to establish a direct relationship between each LC type and LST differences with the mean LST of the study area, independently of other factors. This will enable us to sharpen the LST map and hence highlight areas vulnerable to SUHI with greater precision. Our hypothesis is that daytime SUHIs observed by remote sensing are primarily driven by solar radiant fluxes and not by the air temperature.

## 2. Materials and Methods

### 2.1. Study Area

The primary study area is located in the east of Belgium and covers the city of Liege and its surroundings, spanning 68.4 km$^2$. It extends over approximately 9 km along the west–east axis and 7.5 km along the north–south axis. The climate of the study area is currently temperate without dry season and with warm summer, according to the Köppen–Geiger classification. However, climate change models indicate a possible shift towards temperate climate with hot summer in the future [65]. Considering the heatwaves of 2018 and 2022, summer UHIs have become a concern for the city of Liege, which has led to interests in mitigation solutions.

The city of Liege has a population of approximately 200,000 inhabitants. Its city center is densely built-up, but some districts have a large proportion of tree cover, and the Meuse river runs through the area. It also benefits from the cooling effect of the Ourthe and Vesdre rivers in the South and the Albert Canal in the North.

In addition to the city of Liege, the two largest towns of Wallonia in proximity of Liege are used for an independent testing of the model, namely Namur and Verviers. These towns are however smaller than Liege, with respectively 93,000 and 56,000 inhabitants. Figure 1 displays the land cover around the three urban areas and the respective study areas used for the analyses.

### 2.2. Data

Landsat-8 scenes 198/25 and 197/25 cover the study area. The Landsat 8 TIRS sensor images were used for the LST because of its high spatial resolution and its global coverage of daytime images every 16 days. In this work, 11 daytime images of Collection 1, with less than 10 % cloud cover, were selected on USGS's EarthExplorer (Figure 2). These images belong to the Tier1 category, characterized by consistent georegistration and within prescribed image-to-image tolerances of ≤12 m radial root mean square error (RMSE).

The selected images correspond to the warm weather dates in the 2018–2021 period, including 4 images captured during official heatwaves in Belgium, i.e., periods of minimum 5 consecutive days above 25 degrees Celsius in Uccle, out of which at least 3 days reached

more than 30 degrees Celsius based on the definition from the Royal Meteorological Institute of Belgium. The spatial average of land surface temperature in the city exceeds the maximum air temperature in ninety percent of the cases.

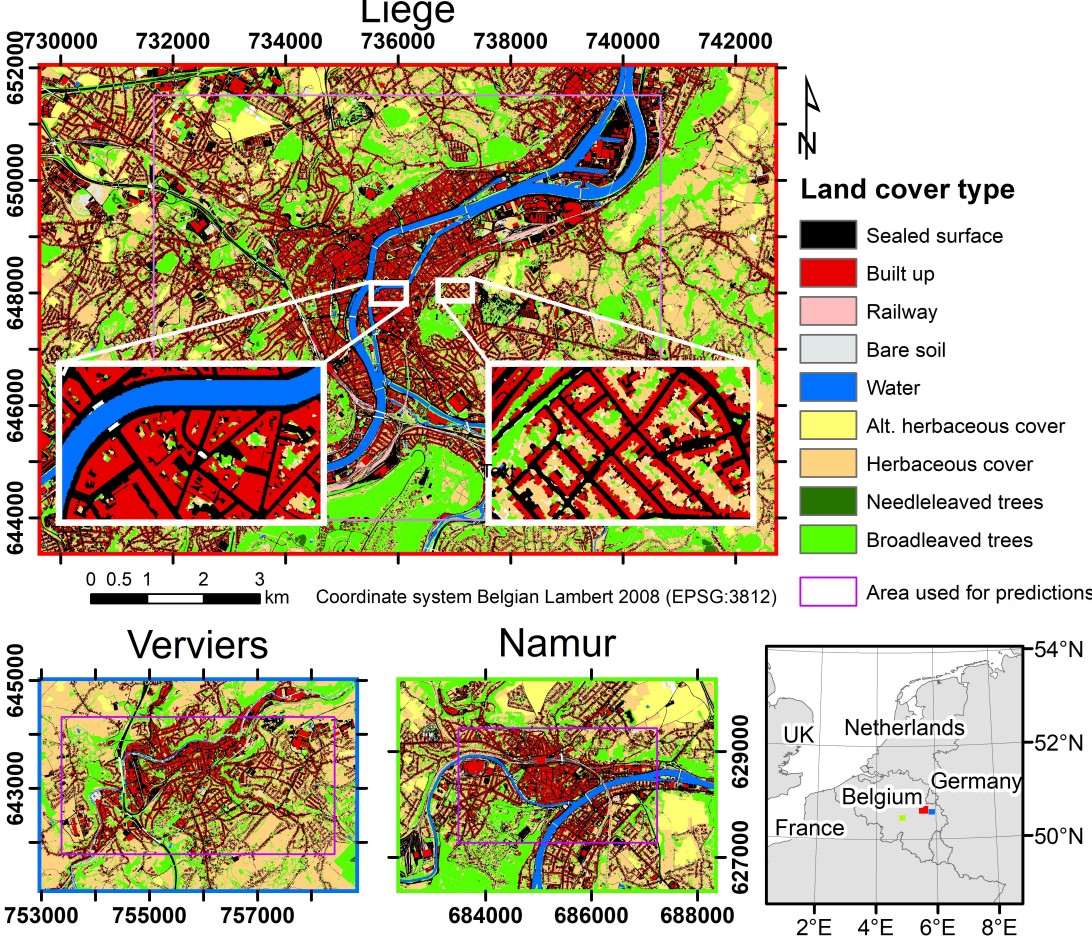

**Figure 1.** Land cover of urban areas used for the calibration (Liège) and the validation (Namur, Verviers) of the model. All are located in Wallonia, the southern region of Belgium.

The land cover map of Wallonia, Walous 2018 [66], was selected for its high spatial resolution (1 m) and high overall accuracy (92.5%), allowing a precise definition of the physical and biological coverage of the land surface. This layer depicts, for the year 2018, the land cover of the whole Walloon territory in 11 primary classes: artificial building, bare soils, water bodies, permanent herbaceous cover, alternating herbaceous cover, needleleaf trees, needleleaf shrub, rail network infrastructure, artificially sealed surfaces (assimilated to "impervious ground"), broadleaf trees and broadleaf shrub. Secondary classes were also described using double labels when two land cover types overlap (e.g., a bridge on top of a river). For these labels, the top label is always taken. In addition, because of the small area covered by shrubs, these two classes were merged with the corresponding "tree" classes (needleleaf trees or broadleaf trees). After merging the original land cover labels, 9 classes remain. However, the class "alternating herbaceous cover" was then split in two classes (crop fields with and without vegetation cover) based on a NDVI threshold of 0.2 derived from the Landsat NIR and red bands. There is indeed a large difference between bare soils and vegetation cover that is not discriminated by a land cover map. The NDVI is computed and applied at each date with the optical bands of the scene used

for LST measurement. In total, there are thus 10 classes at each date, and each class is represented by at least $10^5$ pixels.

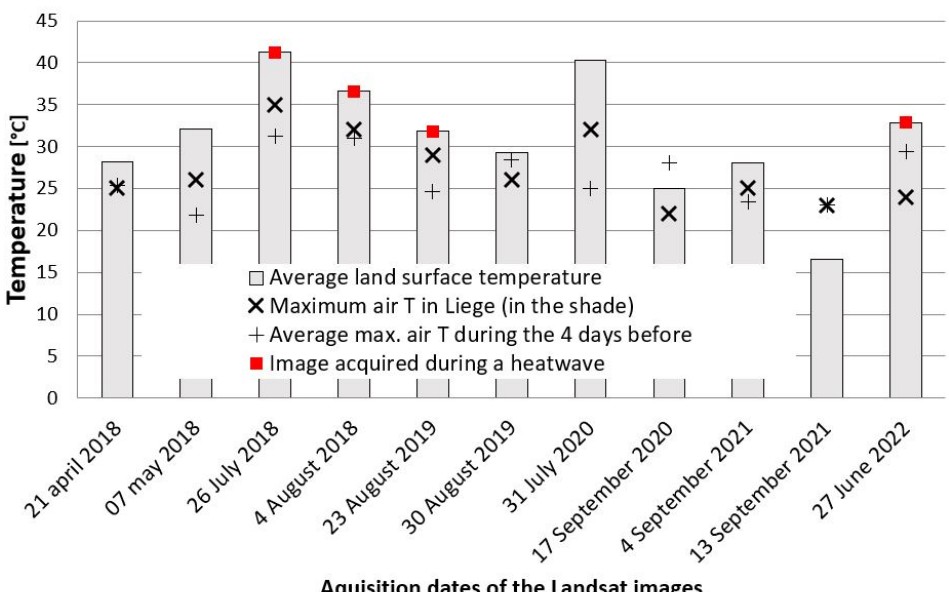

**Figure 2.** Air and surface temperature associated with the images used in the study.

In order to apply the proposed approach to all of Belgium, a dataset with precise information about land cover proportion is necessary. The ecotope dataset of 2015 (the only date available for all of Belgium) was therefore selected. It consists of irregular polygons automatically segmented based on the topography and the land cover [67]. Each polygon includes information about land cover proportions computed based on a 2 m resolution raster of 13 land cover classes (described in [68]).

*2.3. Methods*

The overall approach of the study (Figure 3) is to produce a high-resolution map of relative surface temperatures (ΔLST) in urban areas in order to identify heat sources that could contribute to urban heat islands. This is achieved in several steps. First, Landsat 8 signal is converted to land surface temperature (LST) with state-of-the-art software. Second, we aim to demonstrate the necessity to include the PSF when linking land cover type proportions and thermal radiation. A model with PSF is thus compared with a model without PSF on the same images. Third, we aim to estimate the relative contribution of each land cover type to the emission of thermal radiation. New models are therefore calibrated on the same area at ten different dates, and then the generalized model is validated on two new areas unseen in the calibration phase at an unseen date. Finally, the utility of the linear model is illustrated by mapping the relative LST, and hence locating daylight SUHI regions and their intensity across Belgium. Images were processed with Orfeo Toolbox version 9 [69].

2.3.1. Preprocessing

Landsat TIRS data were processed with LandARTS to derive LST. This application converts the 100 m raw Landsat 8 data in kelvin × 100 LST images at 30 m resolution with an RMSE of 2 K [70]. The kelvins are converted to degree Celsius by subtracting 273.15 degrees. The resulting images are reprojected from UTM 31N/WGS84 to Lambert Belgian 2008 and cropped over the study area.

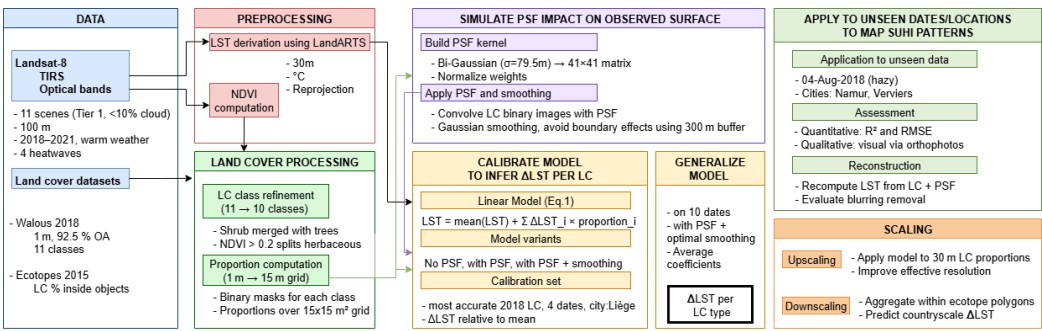

**Figure 3.** Overview of the method.

### 2.3.2. PSF-Based Model and Neighborhood Sensitivity

In order to take into account the PSF of the TIR sensor, a weighting matrix was designed to simulate the effect of the PSF. This matrix imitates the blurring effect on the recorded signal that is stored in each pixel.

The spatial quality of an optical remote sensing instrument is generally obtained by the interaction between the Ground Sample Distance (GSD) of the radiance received by the sensor and the PSF of the instrument's optical system. The GSD is associated to the distance between the centers of adjacent pixels [71]. The PSF of an imaging system refers to a measurement of the amount of blurring formed by the complex interaction between all the components of the imaging system such as the optics, electronics and atmosphere. This implies that the energy received by the imaging system's detector always includes energy from outside the projected footprint of the pixels. In relation to resolution, the PSF therefore expresses the dilution of the signal. Estimating the PSF of an imaging system gives the effective spatial resolution (ESR) of that system [72,73].

Information about the ESR is needed to evaluate the contribution of the neighborhood to each pixel [61]. Wenny et al. (2015) [72] estimated the PSF of Landsat TIRS images using natural targets with sharp edges to determine the edge spread function. The one-dimensional Gaussian parameterization ($\mu = 0$; $\sigma = 79.5$ m) of the Sahara 3 site is used in our paper to derive a bi-dimensional Gaussian PSF (Figure 4). The PSF values are discretized onto a 41 by 41 window of 15 m pixels (half of the Landsat pixel spacing after preprocessing) grid. The weights are normalized by the sum of all values so that the sum of all weights equals 1.

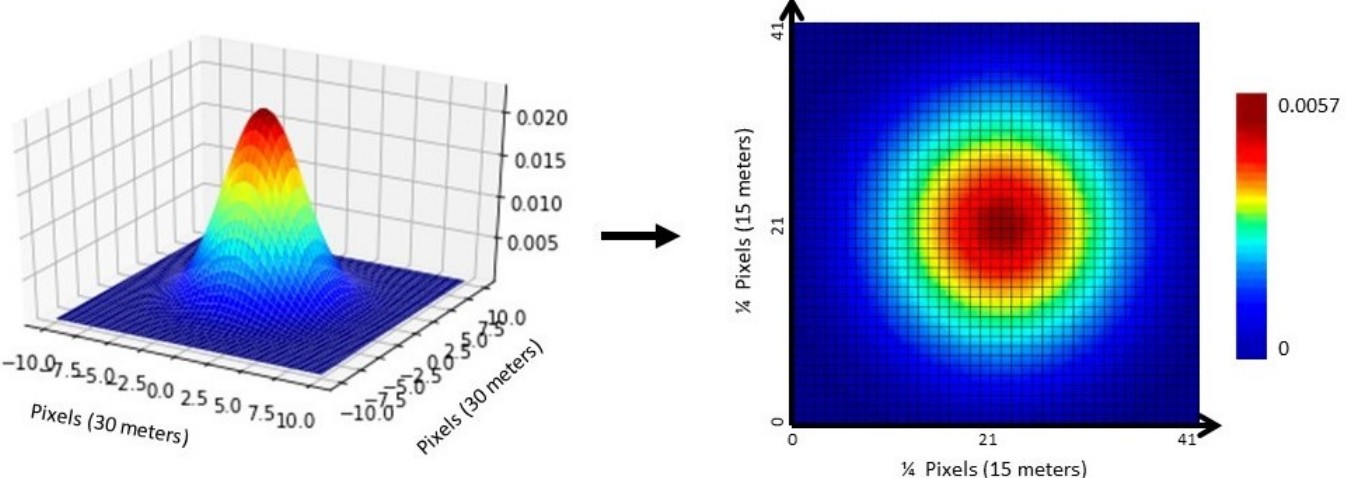

**Figure 4.** Landsat LST point spread function and its resampling to the weighting matrix used to compute pixel contributions.

For each land cover class, a binary map at 1 m resolution is created. The proportion is then computed on a $15 * 15$ m$^2$ grid snapped on the Landsat pixels, based on the 225 1 m$^2$ underlying pixels of the land cover map. In order to obtain the same effective resolution between the land cover data and the LST, the contribution of all land cover pixels is weighted according to their position with respect to the LST measurement. This process is illustrated on Figure 5 in the case of surface water.

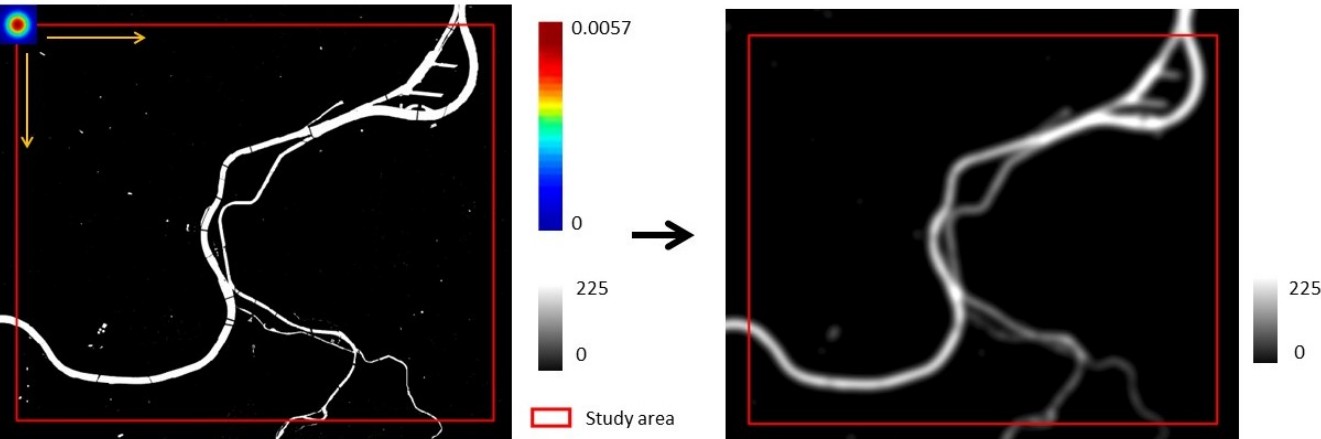

**Figure 5.** Convolution window (upper left corner) applied on a 15 m resolution binary image (**left**) with the PSF matrix to obtain the proportions (**right**) of each land cover in order to estimate their contribution to LST values. The process is illustrated for surface waters in this figure. A buffer around the study area is used to avoid boundary effects.

In addition to the PSF blurring, Gaussian kernels of different standard deviations are applied on each smoothed binary land cover images. This complementary smoothing aims at accounting for other blurring effects such as close-range temperature gradient or image resampling. In all cases, boundary effects are avoided by using a 300 m buffer around the study area for the computation of the PSF and the additional smoothing.

### 2.4. Estimation of the Land Cover Type Effect

The average contribution of each land cover type will be assessed using a system of linear equations. For each of the four dates studied in 2018, one model is calibrated via the proportions of the land cover (i) without taking into account the Landsat TIRS PSF, (ii) taking into account the Landsat TIRS PSF to reach the ESR, and (iii) taking into account the Landsat TIRS PSF and the optimized smoothing kernel.

In order to test the impact of the PSF on the models, only the images of the year 2018 have been used because it is the date with the most accurate land cover information. For the 4 dates, a system of 75,802 equations (see Equation (1)) was built by taking all the pixels of the study area of Liège. The data are centered by subtracting the average LST of the zone. The model is solved using the least squares method. Solving this system of equations results in the estimation of the LST difference to the mean LST for a pixel that was entirely occupied by a single land cover. The system of equations therefore produces the LST difference to the mean LST ($\Delta LST$) for 10 land cover types, namely built-up, rail, sealed surfaces, crop with bare soil, bare soil, water, broadleaf trees, needleleaf trees, permanent herbaceous, and crop with vegetation cover.

$$LST_{Landsat} = mean(LST) + \sum_{i=1}^{10} \Delta LST_i \times Proportion_i \qquad (1)$$

where $LST_{Landsat}$ denotes the LST derived from Landsat LandARTS; $LST_i$ is the average difference to the mean LST of the study area for a pixel entirely occupied by land cover type $i$; and $Proportion_i$ is the share of the pixel affected by land cover type $i$.

After the preliminary analysis, the same method is applied on the other dates and a generalized model is created with the average values of each individual model. With this model, it is possible to produce new rasters estimating the LST of each pixel from the average LST for the study area and the land cover at the studied date.

## 2.5. Accuracy of the Model

For each calibration of the model, the coefficient of determination ($R^2$) and the root mean square error (RMSE) are calculated using all the pixels of the inferred and the reference images. Furthermore, an independent test is performed with the generalized model based on all but one date (the image of 4 August 2018, covered by haze over the city of Liege). The generalized parameters are applied on a new date for two other Walloon towns, namely Namur and Verviers, which were not used during the calibration phase.

In addition to the quantitative assessment, a qualitative assessment is performed on the study area in order to identify potential sources of macroscopic errors in the model. This assessment is performed by a photointerpretation of aerial ortho-images.

## 2.6. Downscaling and Upscaling

Depending on the linear model's performance, it could be possible to reproduce the original Landsat LST images and, in the best case, to reconstruct a perfect sensor image where the blurring is removed. For the reconstruction of the original image, the subpixel land cover proportions are convoluted with the PSF kernel before applying the linear model. However, the model can also be applied on the actual proportions of land cover for each of the 30 m pixels, that is, the total area of the pixels of each land cover type within 30 m pixels divided by 900 m$^2$. By doing so, the effective spatial resolution of the image would be improved, even if the same pixel spacing is maintained. This upscaling is performed with the model optimized for each date.

As the model solely depends on the land cover, it can also be applied on other datasets that have precise information about land cover proportions and compatible land cover classes. For instance, the spatial aggregation on irregular polygons with 2 m resolution boundaries is possible due to datasets from Lifewatch-ERIC available across Belgium: the ecotopes [67]. The use of ecotopes has the advantage of keeping precise land cover boundaries with homogeneous land cover types, which is relevant in our case study because of the limited impact of land cover on its neighborhood. This downscaling is illustrated with the generalized model that provides average LST differences.

# 3. Results

## 3.1. Spatial Resolution

The preliminary step aimed at comparing the results of the models with or without taking the PSF into account. Table 1 demonstrates that the models accounting for the PSF perform better than the models without PSF. Indeed, the average RMSE reduces from 2.35 without PSF to 1.6 with PSF. The coefficient of determination increases on average from 0.57 without PSF to 0.81 with PSF. The better performance of the models with PSF is consistently observed for all dates; therefore, the following steps will focus on the results of the model with PSF. However, the results of the model of 4 August are less certain than the other dates. After a closer look at the image, there appeared to be a residual atmospheric perturbation in the eastern part of the scene. This image was therefore discarded from the other analysis on the city of Liege, but is reused later for testing the generalized model on other cities.

The PSF model performed consistently well for all the dates. The RMSE values ranged from 1.09° to 1.70° with an average of 1.32°. The $R^2$ ranged from 0.876 to 0.959 with an average of 0.91. Different Gaussian kernels were applied on top of the PSF from Wenny et al. (2015) [72] to empirically test other neighborhood's influence. The results showed a non-statistically significant improvement ($5 \times 10^{-9}$ degrees) of the models' RMSE, up to 0.1 pixels (3 m), after which the models' performance deteriorate significantly. The RMSE differences with the RMSE of the PSF models follow a second-order polynomial in the range of the tested values, as shown on Figure 6. These results corroborate the use of the state-of-the-art PSF values.

**Table 1.** Root mean square error (RMSE) and determination coefficient ($R^2$) of the preliminary models obtained with the system of 75,802 equations relating each LST 30 m pixel value of the study area with the respective proportions of each land cover type for the corresponding area computed with and without the PSF effect.

| Dates | RMSE (°C) | | $R^2$ | |
|---|---|---|---|---|
| | With PSF | Without PSF | With PSF | Without PSF |
| 21 April 2018 | 1.32 | 2.09 | 0.85 | 0.59 |
| 7 May 2018 | 1.42 | 2.46 | 0.88 | 0.62 |
| 26 July 2018 | 1.42 | 2.23 | 0.87 | 0.61 |
| 4 August 2018 | 2.11 | 2.57 | 0.66 | 0.47 |
| average | 1.60 | 2.35 | 0.81 | 0.57 |

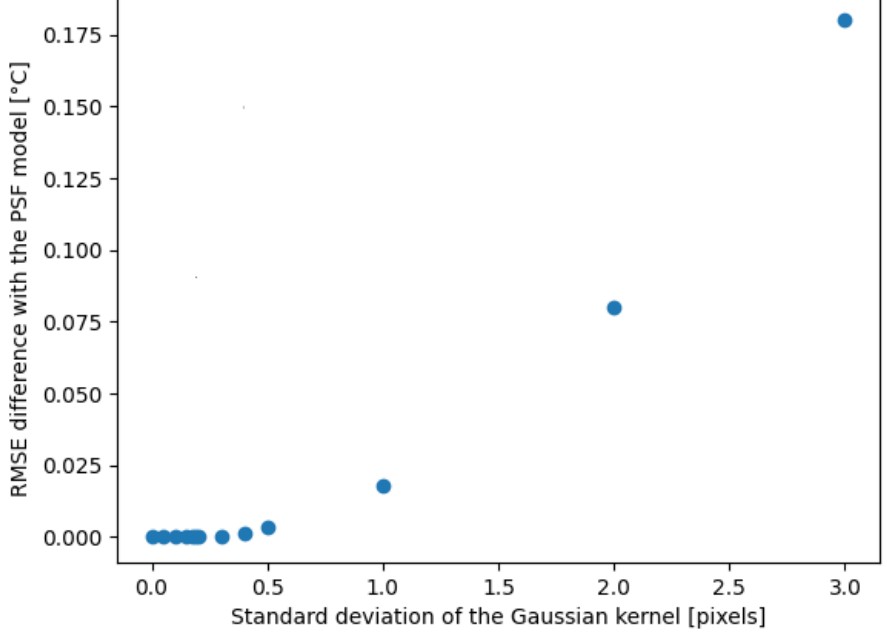

**Figure 6.** Average impact of the additional smoothing with a Gaussian kernel on the RMSE of the models compared with the PSF models on all (10) dates except for August 2018.

### 3.2. Land Cover Types

The parameters of the linear unmixing correspond with the difference in LST caused by each land cover class. Table 1 shows that half of the land cover types induce above-average LST: artificial constructions, alternating herbaceous cover without vegetation, railway network, artificial floor sealing, and bare ground. In contrast, surface waters, broadleaf trees, needleleaf trees, year-round herbaceous cover, and alternating herbaceous cover with vegetation contribute to less-than-average LST. Among the latter, herbaceous covers have less effect on LST than water and trees.

The statistics of Table 2 indicate a small variability of the mean LST differences to the mean despite the different dates and temperature. The confidence interval for water, rail, and needleleaf trees is however larger than 3 °C. On the other hand, the contribution of broadleaf trees and sealed (impervious) surface is very stable (less than 1 °C). Figure 7 suggests that the relative cooling effect of water becomes more important in periods of high temperature, which is when it is most needed.

**Table 2.** Statistics of the contribution of land cover types to the difference in LST for 10 images. The mean value and the 95% confidence interval (CI 95%) are provided for each land cover class.

| Land Cover Type | Mean $\Delta LST$ (°C) | CI 95% (°C) |
|---|---|---|
| Built-up | 8.94 | 1.87 |
| Rail area | 4.62 | 3.90 |
| Sealed surfaces | 3.20 | 0.99 |
| Crop with bare soil | 3.89 | 2.91 |
| Bare soil | 6.70 | 1.22 |
| Water | −7.09 | 3.50 |
| Broadleaf trees | −7.42 | 0.80 |
| Needleleaf trees | −4.62 | 3.74 |
| Permanent herbaceous | −2.07 | 1.66 |
| Crop with vegetation | −3.58 | 1.64 |

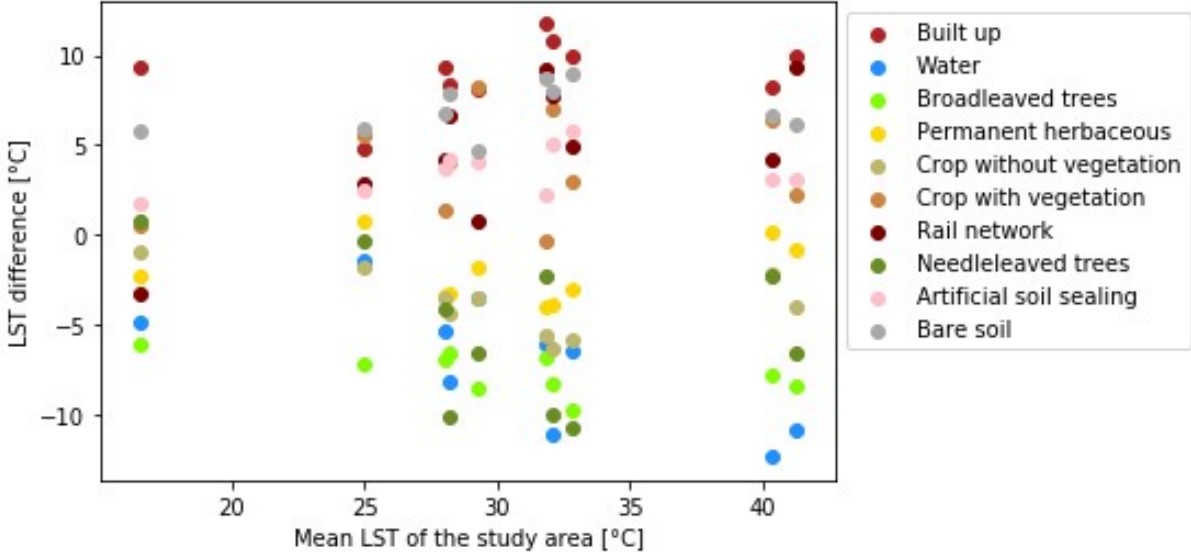

**Figure 7.** Distribution of the LST differences for each land cover type based on the date-specific models according to the mean LST of the study area.

The general linear model was successfully applied to infer LST differences on the two other cities at a date that was not used to build this general model. While the $R^2$ values were smaller than for the calibrated models, it remained high with 0.85 and 0.70 for Namur and Verviers, respectively. This demonstrates that proportions of land cover types can be used to infer the differences in LST with respect to the average LST of a city.

### 3.3. Macroscopic Errors

Over- and underestimations of the LST by the model are observed at the same locations at each dates. A visual inspection based on orthophotos and supported by a roof top classification of Liège [74] highlighted potential sources of errors.

In Liège, most errors are related to large roof tops with unusual building materials. The contribution of metallic and black coating roof tops is underestimated while it is overestimated for glass coatings.

In Verviers, the main errors were linked with (i) the presence of synthetic sport fields and (ii) grasslands with very low NDVI because of the exceptionally dry period in the area at this date. As a result, the LST of these areas was underestimated by the model. There was no macroscopic error in Namur apart from a crop field identified as a grassland in the land cover map.

### 3.4. Applications of the Linear Mixture Model

The parameters of the model can be applied in different ways. First, as shown on Figure 8a, the LST observed by Landsat TIRS can be reproduced using the smoothed land cover proportions and Equation (1). Since the model yields consistently accurate results, it can be used to increase the resolution of the SUHI detection. Indeed, LANDART provides information at a 30 m resolution with a lower effective resolution. By using this model, it is possible to infer the local LST more precisely in terms of spatial resolution. For this purpose, the model calibrated via the land cover proportions with PSF is applied on the effective proportions inside each pixel. This results in a spatially more detailed description of the LST. Figure 8c illustrates the result obtained on 26 July 2018.

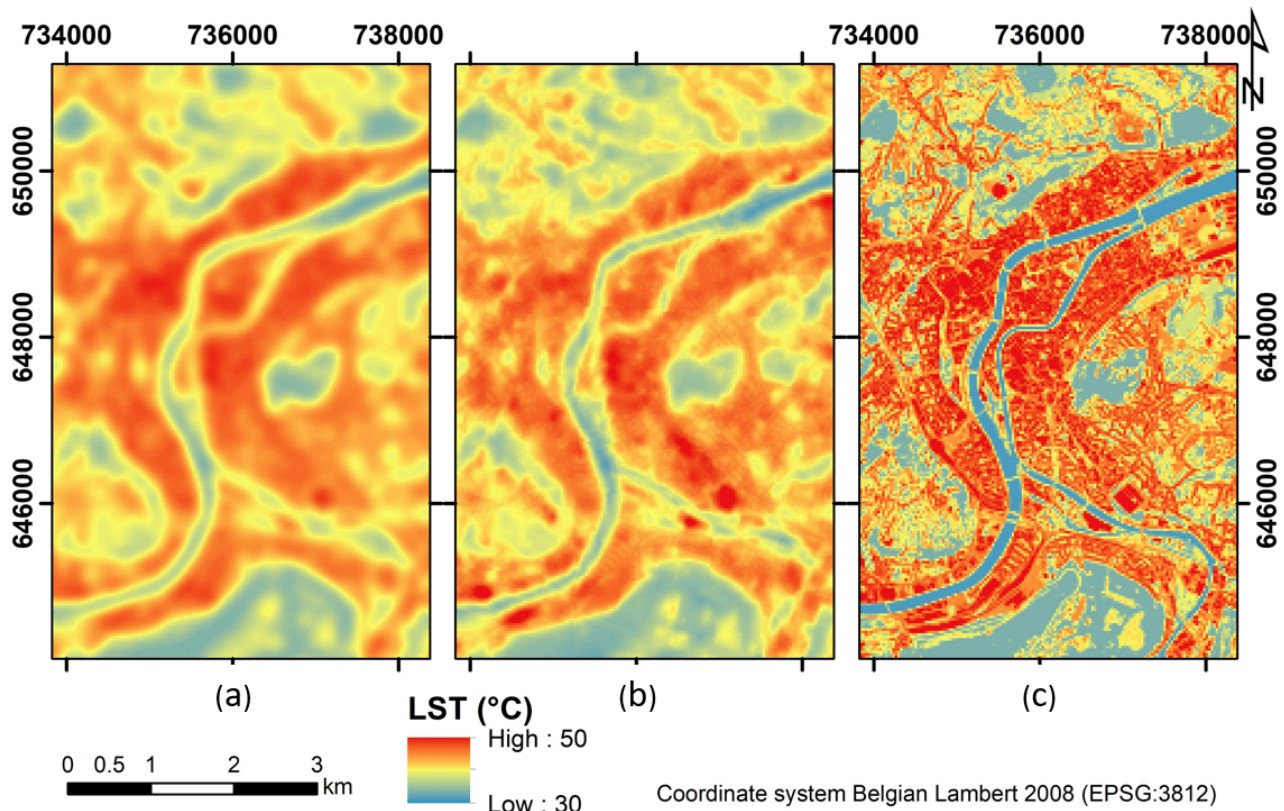

**Figure 8.** Reconstruction of LST maps for 26 July 2018: (**a**) the best fitting model, (**b**) LST observed with Landsat, and (**c**) high-resolution LST obtained by unmixing.

It is also possible to produce LST difference maps to identify potential SUHIs if accurate information about land cover proportion is available. Figure 9 shows such maps for the country of Belgium, where potential SUHIs are displayed in red. The generalized model allows for a direct comparison of cities with a common reference, because the relative contributions of each land cover are not influenced by local context. The continuous values of the mean LST difference provide more details than the Local Climate Zone

classification [75] because the latter do not take into account the type of pervious surfaces mixed with the buildings. Figure 9 highlights several cities with variable areas of greater than average LST. The high resolution of the inference helps to identify Antwerpen as the city of Belgium with the highest SUHIs, despite the fact that the population of Brussels is about twice as large. Despite the major SUHI influence, the CLHI could be quite different in the two cities. Indeed, Antwerpen could benefit from the cooling effect of its river to alleviate atmospheric UHIs when the wind comes from the West, which is often the case. In contrast, the dense center of Brussels is surrounded with areas of moderately high LST values and therefore is less subjected to cooling from air movements.

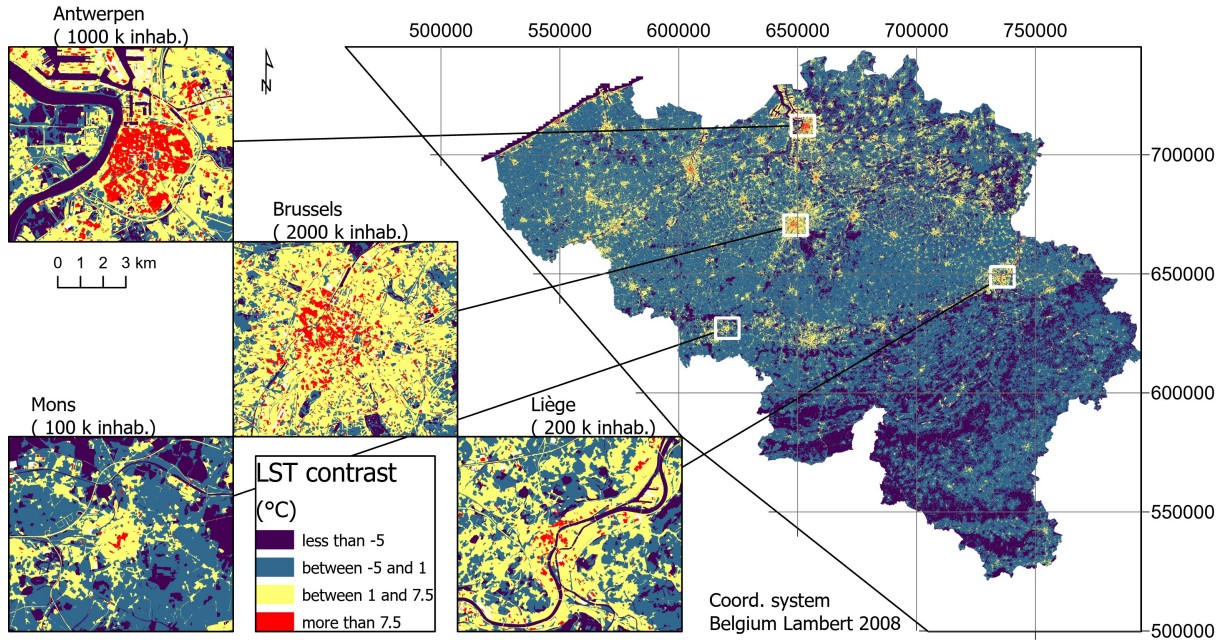

**Figure 9.** Generalized model applied on the ecotopes of Belgium (epoch 2015), with a zoom-in on four cities: Brussels, Antwerpen, Liège, and Mons.

## 4. Discussion

Accounting for the effective spatial resolution of Landsat TIRS demonstrated that land cover types drive the daylight LST differences inside cities, and suggest that other features (such as spatial patterns in the absence of significant 3D effects) have little impact. The thematic precision and the timeliness of the land cover classification are, however, important to compute accurate models. The high quality of the land cover data and their spatial resolution, which is different by one order of magnitude with regard to the LST, is probably the main reason why these results are much better than those of previous studies using land cover-based approaches.

### 4.1. Limitations of SUHI Mapping

Given the excellent results ($R^2$ between 0.88 and 0.96) on the calibration dataset and the good results ($R^2$ = 0.70–0.85 without fine tuning) achieved with the generalized model on an independent date and location, this study confirms a marked contribution of the land cover to the differences in observed LST values. While this relationship was expected, the very large correlation and the low RMSE suggest that the land cover is by far the primary driver of daylight LST, to the point that it could be used to predict daylight LST differences with high confidence.

Among the land cover types, vegetation and water have a lower than average LST, while bare soils, sealed surfaces and built-up areas have a positive contribution. This

information can be used to consolidate climate models because the values of Table 2 can be directly injected in Equation (1) to provide relative LST values of the morning at regional scales. Improving the quality and the spatial resolution of global land cover maps in heterogeneous areas is therefore a key for localizing SUHI. However, the confidence interval on average $\Delta LST$ remains too high to be locally relevant with 2 m pixels: the scope of our model is limited to the sharpening of 30 m pixels (Figure 8c) or aggregating at coarser scales based on precisely known land cover proportions (Figure 9). Furthermore, there are still some limitations that must be further investigated.

First, the qualitative macroscopic error analysis suggests that most errors are due to a lack of thematic precision in the land cover types, and in particular roof top type or specific soil sealing material. It was indeed shown that different roof top and road types had contrasting LST on high-resolution daylight images [76] and that selecting "cool" materials could be used as a passive method to mitigate SUHI [77]. Land cover subtypes enriched by thermal emissivity information would therefore help to further increase the predictive quality of the model, even if the precise estimates of their properties is more challenging if the subtypes rarely occur. Consequently, while the mean values per land cover type already provide enough information for regionally or globally aggregated models, a local analysis still requires the information from remotely sensed thermal data. In addition, the parameters tested in different Belgium cities assume similar building materials in all cities. These results must therefore be used with care in cities with different building materials (e.g., glass). Finally, the physical properties of these materials might change with their humidity, which was assumed to be constant in this study because we focused on warm and dry periods.

Second, the cooling contribution of herbaceous cover and coniferous trees decreased in the last summer months. One hypothesis is that their evapotranspiration was affected by the long lasting drought, while broadleaf trees continued to pump water from the soil because of their deeper root system. For more precision in the model, subclasses could be defined based on the NDVI, which has a proven impact on the LST [59,78]. However, it is not always possible to determine the NDVI of vegetation patches at the same spatial resolution than the land cover map. Indeed, the most likely source of NDVI at the time of Landsat thermal image acquisition is the optical sensor of Landsat itself, with a pixel size of 30 m. This issue could be solved in the near future with the availability of daily images at finer ($\leq$5 m) spatial resolution.

Third, the linear unmixing model was based on land cover proportions computed from a two-dimensional projection. However, the estimation of the contributing surfaces could be improved beyond our PSF model if coregistration errors and three-dimensional structure were taken into account. In locations away from the nadir, the satellite indeed observes part of the sides of tall buildings, which could then hide neighboring objects. Furthermore, local differences in slope and aspect could effect the amount of incident shortwave radiation from the sun, therefore potentially introducing differences that are currently not managed by the proposed model.

Finally, it is worth noting that crop fields follow a seasonal cycle with alternating vegetation cover and bare soil, which have opposite behaviors in terms of LST contrast. It is therefore necessary to take the vegetation cycle into account in case of crop fields in order to avoid large local errors when inferring the aggregated LST difference values based on land cover classes. Because of the limited number of crop parcels in the study areas, the general impact of the crop phenology stage was small. However, it could become large for cities surrounded by (or including) large crop area.

### 4.2. SUHIs Differ from UHIs

The model demonstrates that the PSF of the sensor must be taken into account for the interpretation of the LST based on medium resolution satellites, such as Landsat or Aster. The influence between adjacent land cover types decreases quickly after a few meters, as it is observed in studies with high-resolution thermal images [79]. After removing the blurring effect of the PSF, land cover information explains most local LST differences. Because most building and sealing materials have higher than average LST under the same meteorological conditions, SUHI occur where these materials are used.

Remote sensing provides a snapshot of SUHI under specific conditions. First, most platforms are sun-synchronous, meaning that they systematically collect data at the same time. Second, Landsat image acquisition is only systematic for daylight images and it is impossible to have diurnal and nocturnal images on the same day with a single satellite. Third, images are only usable in (mostly) cloudless conditions. Finally, the near-nadir observation angle mainly captures roofs, tree canopies and horizontal surfaces. Descending (daylight) Landsat images are thus capturing the data from a point of view that is close to 2-D maps at a time where heat sharing is dominated by solar heat fluxes. Those conditions induce major divergences between air and surface temperatures.

Consequently, it is important to properly interpret remotely sensed LST in terms of SUHI and not UHI. LST can directly affect thermal comfort based on radiative transfer in the absence of obstacles to thermal radiation but (i) it depends on air to transmit its heat and (ii) it is much smaller than direct solar radiation. The heat in the air will then travel by convection, which will diffuse it in the neighborhood by circulating air masses. Areas with a large proportion of land cover with high relative LST will therefore be more prone to overheating, but this could be compensated by other land covers of low relative LST, such as water and broadleaf trees, if there are exchanges between air masses. Air mass circulation indeed plays a major role in terms of distribution of the heat, either aggravating or compensating urban heat islands, and this induces complex relationship between air temperature and LST. This could explain why air temperature measured in weather stations and daylight LST do not match well, like it was observed in previous studies [80].

Furthermore, UHI effect peaks during night-time [81] but our results are only valid with daytime SUHI. Night images should therefore be investigated because thermal behavior of the land cover types is markedly different between daytime and night-time. For example, water was a clear heat sink during the day because of its large specific heat, but small water bodies could become a relative source of heat during the night. Previous studies highlighted lower LST contrasts between land cover subtypes at night as air and surface temperature tend to equilibrate [76]. The forthcoming TRISHNA [82] missions, with daytime and night-time acquisitions, should therefore provide more accurate mapping of UHI than daylight TIR images.

In terms of city management, the small distance of influence corroborates that trees and water bodies should be spatially distributed to optimize their area of influence [78]. This study only focused on the mitigation potential of some land cover types in the case of heatwaves, but similar analysis should also consider winter time. The role of vegetation is then still beneficial, but thanks to a relative heating in the city [54]. However, deciduous trees would not contribute as active vegetation during their leaf off period.

### 4.3. Large Area Mapping of SUHIs

The theoretical frequency of acquisition of Landsat images (16 days revisit) is reduced by the occlusion because of cloud cover. Coarser resolution images such as Sentinel-3 can be used to provide a higher revisiting time. However, the spatial resolution of these images is not sufficient to capture SUHI peaks. Previous studies used pansharperning methods to

increase the spatial resolution of the sensor [83,84], but the relationship between visible, near infrared and thermal infrared is not always consistent. For example, a dark metallic roof and water have similar optical characteristics in the visible spectrum, but strongly differ in thermal properties.

Our study showed that, with a well defined reference, it is possible to infer relative LST values on large areas based on land cover data and NDVI time series. However, the model should rely on proportions of pure land cover pixels in order to reach high accuracy. The high resolution (10 m) of currently available global land cover maps, derived by Sentinel-2, is not enough for this purpose because less than 2/3 of the pixels are pure at this resolution in a country like Belgium [85]. Fortunately, higher-resolution data are often available for the largest cities. Alternatively, increasing the thematic precision of 10 m maps with several mosaic classes based on subpixel proportions could also contribute to more accurate models despite a large number of mixed pixels. It then becomes possible to create subclasses of Local Climate Zones based on the proportions of the land cover types.

By extension of the UHI definition, the SUHI is often defined as a difference between the LST in the city and the LST in the surrounding rural area. However, the measurement of SUHI based on a comparison between a city and its surrounding is strongly influenced by (i) the land cover types proportions and (ii) the phenological stages, especially if the rural area is dominated by crop fields. A large proportion of harvested crop fields would indeed increase the LST of the rural area and consequently minimize the indices used to identify SUHI, even if the LST in the city is high. In contrast, a city surrounded by forests more likely shows larger surface heat islands, according to the current definition [86], without necessarily high absolute LST values, simply because of the relatively lower values of the trees. Our results therefore suggest to base the definition of SUHI on one of the land cover types that exhibits a stable phenology when it matters, such as permanent grassland or broadleaf trees, instead of an arbitrary neighborhood distance.

The LST contrast to broadleaf trees allows us to compare different towns without the need to define a "rural area", which is often perceived as an issue in the definition of heat islands. Trees are less sensitive than herbaceous vegetation to limited water availability, and therefore are more resilient to droughts. They are also more consistent than water, which has a high specific heat capacity and multiple origins (sea, lakes, rivers, etc.). In the absence of broadleaf trees, another stable vegetation type could be selected.

## 5. Conclusions

This study demonstrates (i) that the land cover type is the main driver of the differences in LST values observed on daily Landsat images and (ii) that the apparent LST diffusion on Landsat-8 images is primarily due to the point spread function. Consequently, it is possible to infer upscaled 30 m images of heterogeneous urban areas based on land cover information alone. Very high spatial resolution, excellent accuracy, and high level of thematic precision are necessary to achieve locally relevant results, but the contrasting contribution of the different land cover types allows us to highlight general trends at a regional scale. Because of the differences in land cover proportions in the nearby rural areas of each city, our results suggest that the threshold to define a SUHI should be set with respect to the LST average of a reference land cover type in the region, for example, deciduous trees during their vegetation period.

The corollary of the strong correlation between LST and LC proportions is that the influence of air temperature on the differences in remotely sensed LST from morning images is less important than it has been suggested. This further corroborates the difference between Surface Urban Heat Islands and urban heat islands, especially on sunny days when solar radiation dominates other heat fluxes. Further work is necessary to analyze

night LST images, as a different equilibrium is expected when there is no sun. Night images that will become available from high spatio-temporal resolution missions such as TRISHNA or LSTM will play a pivotal role in future studies addressing urban heat islands because daylight images only, primarily measuring the re-emission of the shortwave radiation from the sun, are not sufficient to characterize the diurnal cycle of UHIs.

From a land use planning perspective, our results corroborate other studies, showing that built-up and sealed surfaces reach the highest LST on sunny days, and hence are a major source in the built-up of UHIs. Some more specific points of interest also emerge from our results. On one hand, apart from a river flowing in the city, the best land cover to locally reduce the LST are trees with a deep rooting system, which are more resilient to long heatwaves. On the other hand, the impact of cool building materials should be further investigated and remote sensing could be used to better map them.

**Author Contributions:** Conceptualization, J.R., C.L., A.H. and P.D.; methodology, J.R., A.B., M.D. and P.D.; software, J.R., A.B. and M.D.; validation, J.R. and M.D.; formal analysis, J.R., C.L. and P.D.; data curation, J.R., A.B. and M.D.; writing—original draft preparation, J.R., M.D. and C.L.; writing—review and editing, J.R., A.H. and P.D.; visualization, J.R. and M.D.; supervision, P.D.; project administration, C.L., J.R. and P.D. All authors have read and agreed to the published version of the manuscript.

**Funding:** This study was carried out with the support of the European Space Agency Climate Change Initiative under the contract ESA/No. 4000126564/Land_Cover_cci.

**Data Availability Statement:** The land cover datasets analyzed for this study can be found in the Life-watch Website (https://maps.elie.ucl.ac.be/lifewatch/ecotopes.html, (accessed on 27 March 2024)) and the Landsat images can be found on Earth Explorer (https://earthexplorer.usgs.gov/, (accessed on 27 July 2022)).

**Acknowledgments:** The authors thank the four reviewers for their valuable comments and remarks.

**Conflicts of Interest:** Author Andrew Hartley was employed by MET Office. The remaining authors declare that the research was conducted in the absence of any commercial or financial relationships that could be construed as a potential conflict of interest.

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
