# Peer review of "Land Cover Types Drive the Surface Temperature for Upscaling Surface Urban Heat Islands with Daylight Images"

_remotesensing, doi:10.3390/rs17162815_

Round 1

Reviewer 1 Report

Comments and Suggestions for Authors

This paper explores the techniques for interpolating or weaving land-surface temperature (LST) data from satellite observation. The motivation is to gain a better interpretation of heterogeneous LST data by upscaling/downscaling, for applications to urban research. The key analysis is performed over several cities in Belgium. The results are potentially useful. The manuscript is clearly written. I do have a few concerns:

(1) Although the authors mentioned a better understanding of urban heat island (which is more pronounced at nighttime) and diurnal cycle of LST as a motivation, only daytime observation is analyzed. It would be useful to comment on whether (and how) the methodology developed can be applied to nighttime data. Also, how would the results be used to improve the analysis of the diurnal cycle?

(2) Although Eq. (1) looks impressive with so many sub-categories of land cover, I wonder if some of the categories are just rare occurrences that contribute very little in the formula. For example, from the maps in Fig. 1, I could hardly identify the pixels for “Railway network”. It would be useful to comment on the relative contribution of the sub-categories to the formula in Eq. (1). Moreover, whether it is essential to have all 10 categories in the equation.

(3) The inset in Fig. 5 gives the impression that the analysis can discern the root-mean-square error of LST to the accuracy of 0.0000001 degree C. I believe this is false (considering the degrees of freedom needed to achieve this). Moreover, for applications, such an accuracy is not meaningful nor necessary. Also, note that Table 2 only shows the RMSE of LST to the accuracy of 0.01 degree C (which is more reasonable). I would suggest modifying Fig. 5, perhaps just removing the inset.

(4) The numbers (644000, 646000, etc.) in the labeling of the borders of the maps in Fig. 1 are not properly explained.

Author Response

This paper explores the techniques for interpolating or weaving land-surface temperature (LST) data from satellite observation. The motivation is to gain a better interpretation of heterogeneous LST data by upscaling/downscaling, for applications to urban research. The key analysis is performed over several cities in Belgium. The results are potentially useful. The manuscript is clearly written. I do have a few concerns:

(1) Although the authors mentioned a better understanding of urban heat island (which is more pronounced at nighttime) and diurnal cycle of LST as a motivation, only daytime observation is analyzed. It would be useful to comment on whether (and how) the methodology developed can be applied to nighttime data. Also, how would the results be used to improve the analysis of the diurnal cycle?

 > We are sorry about the confusion between “diurnal event” (something occurring during daylight) and “diurnal cycle” (any pattern that recurs every 24 hours as a result of one full rotation of the Earth around its axis). In our paper, we wanted to say “an image that is acquired during daylight”. We replaced “diurnal” with “daylight” to avoid the confusion, and the title was adjusted. The discussion about night images has been updated. In order to study the diurnal cycle, geostationary satellite would be needed, but their spatial resolution is much coarser (e.g. MSG has a resolution of 3 km). MODIS offers frequent mid-nightime images at 1 km resolution and some HR nighttime images can be tasked (Aster, Landsat) but we didn’t have night images of the area. That being said, the same methodology could be applied on nighttime images, but we expect lower correlation between land cover categories and LST on nighttime images because the heat fluxes are different during the night. For example, we would expect water bodies to because heat sources because of their large specific heat capacity. Therefore we want tried to insist on the fact that our study is (unfortunately) only relevant for daylight images, and we looking forward to new missions such as TRISHNA to be able to analyze nighttime behaviors.

(2) Although Eq. (1) looks impressive with so many sub-categories of land cover, I wonder if some of the categories are just rare occurrences that contribute very little in the formula. For example, from the maps in Fig. 1, I could hardly identify the pixels for “Railway network”. It would be useful to comment on the relative contribution of the sub-categories to the formula in Eq. (1). Moreover, whether it is essential to have all 10 categories in the equation.

 > The equation was modified with indices to make it more readable. As mentioned in section 4.1, we expect to improve the quality of our model if we could have even more categories. The errors that we observed were most of the time linked with marginal categories that were not present in our map (e.g. synthetic sport field, glass covered building…) and we wish that we could increase this. Of course, the confidence interval for these rare classes is larger than more frequent classes (as can be seen on Table 2), therefore extending the area to other cities should be done to improve the model. However, there are areas in the cities were the “rare” categories make a huge difference (for example, the area around the central station of Liège has a large concentration of rails, therefore removing this category would underestimate the heat near the station.)

(3) The inset in Fig. 5 gives the impression that the analysis can discern the root-mean-square error of LST to the accuracy of 0.0000001 degree C. I believe this is false (considering the degrees of freedom needed to achieve this). Moreover, for applications, such an accuracy is not meaningful nor necessary. Also, note that Table 2 only shows the RMSE of LST to the accuracy of 0.01 degree C (which is more reasonable). I would suggest modifying Fig. 5, perhaps just removing the inset.

 > Thank you for your comment. The inset was removed as suggested. That being said, we considered that showing that there is no significantly significant differences for small differences in terms of PSF width is an interesting information on the sensitivity of our model. Furthermore, contrary to table 1, we are looking at the pairwise difference of values with similar PSF and not the absolute values. The distribution of the residuals therefore had a very small variance because of the spatial correlation of the predictions and the degrees of freedom are large (75802), so even a small difference could be statistically significant (it is not the case here, therefore we agree that we should not have zoomed so much).  This has been clarified in the text.

(4) The numbers (644000, 646000, etc.) in the labeling of the borders of the maps in Fig. 1 are not properly explained.

> These numbers are cartographic coordinates in m, the unit of EPSG:3812 coordinate system indicated on the figure. EPSG:3812 is the official coordinate system of Belgium: Belgian Lambert 2008 projection.

Reviewer 2 Report

Comments and Suggestions for Authors

This manuscript conducts a downscaling study of LST data based on high spatial resolution LC data. The study considers the influence of PSF on LST downscaling and improves the situation where previous studies lacked consideration of PSF, significantly enhancing the results of LST downscaling. The theme of the paper is of great value, but minor revisions are still required.

In the introduction section (Line 79-82), the authors are expected to elaborate in more detail on how PSF affects the LST downscaling effect, including the underlying principles and the shortcomings of existing research.

The authors need to further demonstrate the enhancement of LST downscaling by considering PSF in the manuscript. In addition to accuracy metrics, it is recommended that the authors attempt to compare the effects of the two downscaled images to more intuitively illustrate the advantages of the research method in this study.

It should be noted that for land surfaces with complex three-dimensional structures such as urban areas, PSF is not always as regular as depicted in Figure 3. This may have a significant impact on the downscaling results, which needs to be carefully discussed in the manuscript.

Author Response

This manuscript conducts a downscaling study of LST data based on high spatial resolution LC data. The study considers the influence of PSF on LST downscaling and improves the situation where previous studies lacked consideration of PSF, significantly enhancing the results of LST downscaling. The theme of the paper is of great value, but minor revisions are still required.

In the introduction section (Line 79-82), the authors are expected to elaborate in more detail on how PSF affects the LST downscaling effect, including the underlying principles and the shortcomings of existing research.

  • To the best of our knowledge, studies addressing the downscaling of thermal images using land cover data did not take the PSF into account. In other words, they matched the content of pixels with pixel values that are markedly influenced by a neighbourhood exceeding the “square” defined by the pixel on the ground. Several studies in similar context with optical data proved that the PSF must be taken into account. Therefore we also suggest to do it in our study. This was clarified in the introduction.

The authors need to further demonstrate the enhancement of LST downscaling by considering PSF in the manuscript. In addition to accuracy metrics, it is recommended that the authors attempt to compare the effects of the two downscaled images to more intuitively illustrate the advantages of the research method in this study.

  • Figure 3 illustrate the effect on the downscaling with the reconstruction of the signal and the extrapolation of the mean at 2 m resolution. Extrapolating the mean values based on the model that does not take PSF into account is visually similar. We therefore believe that the best visualization is Figure 5 with its increasing RMSE and that the significative differences with and without PSF of table 1 are obvious.

It should be noted that for land surfaces with complex three-dimensional structures such as urban areas, PSF is not always as regular as depicted in Figure 3. This may have a significant impact on the downscaling results, which needs to be carefully discussed in the manuscript.

  • This is a good remark, thank you. With the relatively low height of building in the cities, the satellite viewing angles close to nadir and the coarse resolution of the sensor, the impact on our model would remain limited. But it would indeed become necessary to better estimate the contributing area in cities with high skyscrapers. This issue was added in the fifth paragraph of section 4.1

Reviewer 3 Report

Comments and Suggestions for Authors

The article uses a linear mixing model based on land cover maps to attempt to improve the resolution of LST images from Landsat satellites to 2 m. In order for the article to be published after revision, we hope the authors will respond to the following questions.

  • Please reorganize the abstract of the article, reduce the background information, and directly state the purpose of the article, the main results, and the main conclusions.
  • The existing literature review is insufficient and should be supplemented with commonly used methods for LST inversion and LST image scale conversion.
  • In the sixth line of the abstract, the article argues that convection is the primary means of heat exchange between the Earth's surface and space. However, it is widely accepted that the primary means of heat exchange between the Earth's surface and the atmosphere is that the surface absorbs shortwave radiation from the sun and then releases heat in the form of longwave radiation, while the atmosphere absorbs the longwave radiation and warms up. The article's current statement is neither rigorous nor consistent with mainstream academic views.
  • In the methods section, the article should include a technical roadmap or article structure diagram. It should also explain the purpose of each step and how it affects the next step, rather than simply describing what was done.
  • We note that the LST inversion method LANDART used in the article only refers to a single remote sensing indicator, NDVI, for surface emissivity. However, the article aims to explore the impact of different surface objects on LST inversion. Therefore, when calculating surface emissivity, different surface objects should use different surface emissivity values.
  • The article should use a three-line table.
  • The conclusion of the article is confusing. The main contribution of the article is to propose a method for downscaling LST images, rather than proving that surface features have a significant impact on LST. The impact of surface features and the point spread function of LST are both assumptions on which the downscaling method is based. The current conclusion is circular reasoning.

Author Response

The article uses a linear mixing model based on land cover maps to attempt to improve the resolution of LST images from Landsat satellites to 2 m. In order for the article to be published after revision, we hope the authors will respond to the following questions.

Please reorganize the abstract of the article, reduce the background information, and directly state the purpose of the article, the main results, and the main conclusions.

  • The abstract has been reorganized

The existing literature review is insufficient and should be supplemented with commonly used methods for LST inversion and LST image scale conversion.

  • Literature review has been consolidated

In the sixth line of the abstract, the article argues that convection is the primary means of heat exchange between the Earth's surface and space. However, it is widely accepted that the primary means of heat exchange between the Earth's surface and the atmosphere is that the surface absorbs shortwave radiation from the sun and then releases heat in the form of longwave radiation, while the atmosphere absorbs the longwave radiation and warms up. The article's current statement is neither rigorous nor consistent with mainstream academic views.

  • You are completely right. We wanted to stress the difference between radiation (primary heat exchange during the day) and convection (which is relatively more important at night when there are no more incident shortwave) but it was mixed up in our abstract. This sentence was removed when we reorganized the abstract.

In the methods section, the article should include a technical roadmap or article structure diagram. It should also explain the purpose of each step and how it affects the next step, rather than simply describing what was done.

  • A technical roadmap has been added

We note that the LST inversion method LANDART used in the article only refers to a single remote sensing indicator, NDVI, for surface emissivity. However, the article aims to explore the impact of different surface objects on LST inversion. Therefore, when calculating surface emissivity, different surface objects should use different surface emissivity values.

  • We used LANDART because it is state-of-the-art and their calibration results were good (less than 2 degrees RMSE), even if we agree that it is not fully relevant. However, the differences in LST that we observed are composed of a combination of the temperature of the surface and the emissivity, which cannot be disentangled without a comprehensive emissivity database of each object type or the precise temperature of these objects.

The article should use a three-line table.

  • Tables have been updated

The conclusion of the article is confusing. The main contribution of the article is to propose a method for downscaling LST images, rather than proving that surface features have a significant impact on LST. The impact of surface features and the point spread function of LST are both assumptions on which the downscaling method is based. The current conclusion is circular reasoning.

  • Without areas of homogenous land cover of the size of the satellite observation in urban areas, we had no choice but work with a subpixel model to determine the impact of surface feature on LST. As we now mention in the introduction, it has already been proven with optical data that using PSF is the right thing to do. Because it hadn’t been done with thermal data to the best of our knowledge, we had to demonstrate it in the first experiment of our paper. We then consider that the high correlation between LST and the proportions of land cover type proves that surface feature have a significant impact on LST, more than any other feature given that we explain more than 90% of the variability. Our 95% confidence intervals around the estimated mean LSTdif values is around 2.2° on average (rails and coniferous trees are less precise because of their low frequency in the urban areas). The feasibility of the downscaling is a consequence of this significant impact, and we are confident that the high RMSE on unseen data is a proof that the average LSTdif values are reliable estimates.

Reviewer 4 Report

Comments and Suggestions for Authors

This manuscript presents a compelling and technically solid study on the relationship between land cover (LC) types and land surface temperature (LST) patterns in urban areas. The manuscript demonstrates strong potential for publication, but a few critical improvements are required prior to acceptance:

A. Major comments:

  1. The manuscript would benefit from careful re-reading to eliminate a few editorial inconsistencies. For instance, Lines 111–114 (Section 2.2) repeat similar information about the Landsat 8 TIRS sensor’s resolution and revisit frequency. Additionally, there is a minor contradiction: it first mentions an 80m resolution and shortly after states 100m, which could confuse readers unfamiliar with the sensor’s technical specifications.
  2. Another similar editorial issue occurs around Lines 198–201, where the distinction between models with and without PSF is discussed multiple times across different sections without new insights. A tighter consolidation of these discussions would improve the paper’s focus.
  3. While the methodology rightly emphasizes the importance of PSF correction and LC precision, it neglects to discuss the critical preprocessing step of coregistering thermal and land cover datasets. Precise spatial alignment is vital given the scale mismatch and the convolution-based mixing model. The omission of a description of the image coregistration accuracy, error, or validation is a gap that should be addressed explicitly in Section 2.3.1 or 2.3.2.
  4. The extension of the study to the national level (Section 3.4 and Figure 8) is ambitious but poorly substantiated in the main text. The model's application at a broader scale (e.g., Antwerp, Brussels) is only qualitatively discussed, and there’s minimal analysis of model limitations under different urban morphologies, climates, or data inconsistencies. The recommendation is either defer this national-scale application to a follow-up study or substantially expand the analysis of these large-scale results, particularly regarding spatial error, uncertainty, and heterogeneity.
  5. The assumption of a linear relationship between land cover proportions and LST is reasonable and supported by high R² values; however, the manuscript should at least mention and briefly test whether a non-linear or machine learning model (e.g., random forest, kernel ridge regression) might further improve predictive performance or capture localized variation. Even if linear models remain preferable for interpretability, a short comparative analysis or discussion would strengthen the justification for the linearity assumption.
  6. While the cities analyzed (Liège, Namur, Verviers) are relatively flat, it would be prudent to discuss topographic influence (e.g., slope, aspect, altitude) on LST patterns. These factors can influence solar exposure, especially in non-flat urban areas. The authors briefly mention elevation when describing ecotopes, but no explicit analysis is included. A short discussion in Section 4 (e.g., under limitations) is warranted.

B. Minor Comments:

  1. Line 137: "measure" should be corrected to "measurement."

  2. Line 245: “downscaling is performed with the generalized model...” — consider clarifying whether this includes validation on known data or is purely illustrative.

  3. Table 2: Use consistent decimal precision (e.g., 2.0 instead of 2).

  4. Add a brief explanation in the conclusion of how future missions like TRISHNA will resolve current temporal limitations of SUHI estimation.

Author Response

This manuscript presents a compelling and technically solid study on the relationship between land cover (LC) types and land surface temperature (LST) patterns in urban areas. The manuscript demonstrates strong potential for publication, but a few critical improvements are required prior to acceptance:

  1. Major comments:
  1. The manuscript would benefit from careful re-reading to eliminate a few editorial inconsistencies. For instance, Lines 111–114 (Section 2.2) repeat similar information about the Landsat 8 TIRS sensor’s resolution and revisit frequency. Additionally, there is a minor contradiction: it first mentions an 80m resolution and shortly after states 100m, which could confuse readers unfamiliar with the sensor’s technical specifications.
  • Thank you for this remark, there was indeed an error that has been corrected. The sampling distance is 100m.
  1. Another similar editorial issue occurs around Lines 198–201, where the distinction between models with and without PSF is discussed multiple times across different sections without new insights. A tighter consolidation of these discussions would improve the paper’s focus.
  • This has been checked, but I don’t understand where I should avoid the repetition around 198-201. Could you please add more details.
  1. While the methodology rightly emphasizes the importance of PSF correction and LC precision, it neglects to discuss the critical preprocessing step of coregistering thermal and land cover datasets. Precise spatial alignment is vital given the scale mismatch and the convolution-based mixing model. The omission of a description of the image coregistration accuracy, error, or validation is a gap that should be addressed explicitly in Section 2.3.1 or 2.3.2.
  • Information about the registration error of the Landsat images has been added. However, we trust this registration because there are many reference data in Europe that helped the geometric correction. And we wouldn’t have good results if the images were not well coregistered.
  1. The extension of the study to the national level (Section 3.4 and Figure 8) is ambitious but poorly substantiated in the main text. The model's application at a broader scale (e.g., Antwerp, Brussels) is only qualitatively discussed, and there’s minimal analysis of model limitations under different urban morphologies, climates, or data inconsistencies. The recommendation is either defer this national-scale application to a follow-up study or substantially expand the analysis of these large-scale results, particularly regarding spatial error, uncertainty, and heterogeneity.
  • We will defer this national scale application if you ask for it again in a next revision, but please consider that we’ve spent a lot of effort to rigorously validate our model (2 unused areas and an unused image) therefore we trust the prediction on other cities. Of course, there are more skyscraper in Brussels than in Liege and our model could be further improved (as discussed), but we expect that the spatial aggregation per polygons nullifies spatial errors and drastically reduces uncertainties (the heterogeneity is managed by the model itself as we know precisely the land cover proportion in all polygons).
  1. The assumption of a linear relationship between land cover proportions and LST is reasonable and supported by high R² values; however, the manuscript should at least mention and briefly test whether a non-linear or machine learning model (e.g., random forest, kernel ridge regression) might further improve predictive performance or capture localized variation. Even if linear models remain preferable for interpretability, a short comparative analysis or discussion would strengthen the justification for the linearity assumption.
  • Our linear model is physically sound (the amount of energy emitted by a surface is directly proportional to its area and it adds up), and non linear models are more likely to overfit, even if we have a large number of points. Considering previous studies, our main innovation is to include the PSF (and also, we had the chance to have a very high resolution land cover, but this is not really an innovation). As you mention, our R² values are good : a linear relationship is considered to be “very strong” if the absolute R value is between 0.9 and 1 (in our case, the mean R² is 0.91, so the absolute R is 0.95).
  1. While the cities analyzed (Liège, Namur, Verviers) are relatively flat, it would be prudent to discuss topographic influence (e.g., slope, aspect, altitude) on LST patterns. These factors can influence solar exposure, especially in non-flat urban areas. The authors briefly mention elevation when describing ecotopes, but no explicit analysis is included. A short discussion in Section 4 (e.g., under limitations) is warranted.

> This has been added to the discussion.

  1. Minor Comments:
  1. Line 137: "measure" should be corrected to "measurement."
  • done
  1. Line 245: “downscaling is performed with the generalized model...” — consider clarifying whether this includes validation on known data or is purely illustrative.
  • We replaced “is performed” with “is illustrated”
  1. Table 2: Use consistent decimal precision (e.g., 2.0 instead of 2).
  • done
  1. Add a brief explanation in the conclusion of how future missions like TRISHNA will resolve current temporal limitations of SUHI estimation.
  • They will capture night and day images in the TIR globally, unlike Landsat

Round 2

Reviewer 3 Report

Comments and Suggestions for Authors

This paper proposes a method for LST downscaling  from Landsat images based on high-resolution LULC images.These areas still need to be revised before publication.

1)The methods section lacks clear flowcharts.

2)The article proposes a downscaling method based on machine learning and image technology, which can indeed improve the accuracy of LST to a 2m scale. However, due to the lack of high-resolution LST images or actual measurement data at a 2m scale in real-world conditions, the article cannot prove that its downscaling method is more consistent with reality. This is what the authors need to pay attention to.

Author Response

Dear reviewer 1,

A flowchart was added as requested. 

We fully agree that our results cannot be used at 2m resolution, which we didn't. The two meter resolution is necessary to obtain a precise estimate of the land cover proportion in urban areas, which are often very heterogeneous in terms of land cover (on most 10 m land cover maps of the city of Liege, 80% of the pixels would be "urban" with some "forest" pixels in a few large parks and some "water" pixels for the river). But our model only extract the average LST of land cover type, not the specific values at 2 meter resolution (which might change due to aspect, material difference...). Two sentences have been added in the discussion about the limitations to clarify this point: "However, the confidence interval on average ∆LST remains too high to be locally relevant with 2-meter pixels: the scope of our model is limited to the sharpening of 30-m pixels or aggregating at coarser scales based on precisely known land cover proportions."

Reviewer 4 Report

Comments and Suggestions for Authors

The authors responded at all demands and comments. 

Author Response

Thank you, your comments were useful.